# Efficient Prealignment of CT Scans for Registration through a Bodypart Regressor

**Hans Meine**[1,2]                                            MEINE@UNI-BREMEN.DE

**Alessa Hering**[2,3]                        ALESSA.HERING@MEVIS.FRAUNHOFER.DE

[1] *University of Bremen, Medical Image Computing Group*

[2] *Fraunhofer Institute for Digital Medicine MEVIS*

[3] *Diagnostic Image Analyse Group, Radboud UMC, Nijmegen, Netherlands*

## Abstract

Convolutional neural networks have not only been applied for classification of voxels, objects, or images, for instance, but have also been proposed as a bodypart regressor. We pick up this underexplored idea and evaluate its value for registration: A CNN is trained to output the relative height within the human body in axial CT scans, and the resulting scores are used for quick alignment between different timepoints. Preliminary results confirm that this allows both fast and robust prealignment compared with iterative approaches.

**Keywords:** SSBR, self-supervised bodypart regressor, registration, prealignment

## 1. Introduction

Bodypart recognition is an interesting task that has many potential benefits for workflow improvements. For instance, it may be used for data mining in large PACS systems, for triggering automatic preprocessing or analysis steps of relevant body regions, or for offering optimized viewer initializations for human readers during a particular kind of study.

The recent advent of convolutional neural networks in medical image computing has also led to several applications for bodypart recognition: Yan et al. (2016) and Roth et al. (2015) trained CNN that classified axial CT slices into one of 12 and 5 manually labeled body parts, respectively. While such an assignment is reasonable on a scan level, it has the downside that slices in transition regions (e.g. thorax / abdomen) cannot be uniquely assigned and that the information is rather coarse. Hence, later works (Zhang et al., 2017; Yan et al., 2017) posed the problem as a regression of the relative body height, which allows more fine-grained region identification and does not suffer from ambiguities. While Zhang et al. (2017) used manually labeled anatomical landmarks for calibration, Yan et al. (2017) suggested a novel training approach that no longer needs *any* manual annotation, but can be trained on a large number of unlabeled transversal CT scans. The latter method was introduced for mining RECIST measurements (Yan et al., 2018) and is used in this work.

Registration of CT volumes has also been recently approached with CNN (Eppenhof et al., 2019; Hering and Heldmann, 2019; de Vos et al., 2019). Most of these approaches focus on deformable registration. However, a full registration pipeline typically consists of several steps, starting with a coarse prealignment. This prealignment is particularly challenging when the two scans cover different regions or have only a small overlap.

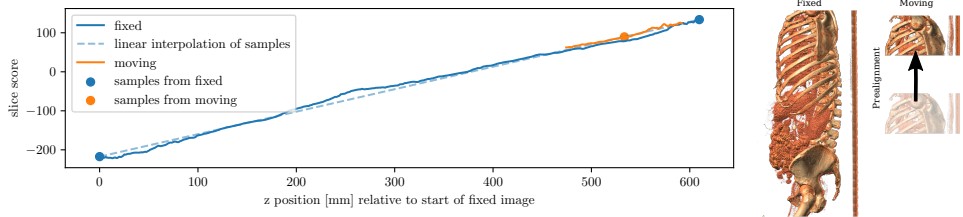

Figure 1: Slices scores of a random image pair, preligned based on three samples

## 2. Materials and Methods

For the experiments described in the following, we used a dataset of 1475 thorax-abdomen CT scans in transversal orientation from 489 patients acquired at Radboud UMC, NL in 2015. After a patient-level split into training and test sets, we trained a bodypart regressor on a subset of 1035 volumes (from 326 patients) that had at least 300 slices each, comprising a total of 670.986 slices, leaving a set of 440 volumes with 284.258 slices for testing. These test volumes result in 277 intra-patient registration pairs. To generate more challenging cases, only a subvolume of the moving image is used for the registration. For this purpose, a random start slice is uniformly sampled between the first and the *end-100*th slice. Additionally, the number of slices is uniformly chosen with at least 20 slices up to the whole volume.

The self-supervised bodypart regressor (SSBR (Yan et al., 2018), aka UBR (Yan et al., 2017)) is based on a modified VGG-16 network that uses global average pooling and a single dense layer after the convolutional basis to output a single score for each slice, resampled to $128 \times 128$ voxels. The key ingredient is the loss function, which consists of two parts: Given a batch of 32 stacks of $m = 8$ equidistant slices, the loss $L_{\mathrm{SSBR}} = L_{\mathrm{order}} + L_{\mathrm{dist}}$ is a sum of a term $L_{\mathrm{order}} = -\sum_{i=0}^{m-2} \log h\left(s_{i+1} - s_i\right)$ that penalizes non-increasing scores within the stack and a term $L_{\mathrm{dist}} = \sum_{i=0}^{m-3} g\left(\Delta_{i+1} - \Delta_i\right)$ for achieving equal differences $\Delta_i = s_{i+1} - s_i$ between scores of equidistant slices ($h$ is the sigmoid function and $g$ is the smooth L1 loss). By sampling random stacks of equidistant slices with varying positions and inter-slice spacings, the network learns to output linearly increasing scores via short and long distance penalties. Absolutely no manual annotation is necessary, not even landmarks.

For registration prealignment based on the regression scores, we devised two methods. The first, extremely fast approach is to compute the score of the first and last slices of the fixed image and the score of the center slice of the moving image. This allows us to estimate the relative position of the center of the moving image within the fixed image for prealignment, based on scoring just three slices (cf. Figure 1). The second method computes the scores of all slices, resampling to a common slice spacing, and then computes the best match by shifting one score curve with respect to the other, identifying the position with the lowest $\ell_1$ norm of the overlapping parts (Figure 2).

We compare these SSBR prealignments with a brute force grid search method named FASTA (Fast Translation Alignment) which evaluates a difference measure (here SSD, the squared $\ell_2$ norm of the difference image) on a grid of possible translations. Finer grids allow for more precise translation estimation at the expense of increased computational cost. For

Table 1: Scoring results for all methods with the following categories: 1: very good alignment, 2: good alignment, 3: correct body region, and 4: failure.

| Method | Mean Score | 1 | 2 | 3 | 4 |
|---|---|---|---|---|---|
| FASTA | 1.6 | 188 | 41 | 17 | 31 |
| SSBR fast | 1.7 | 127 | 113 | 23 | 14 |
| SSBR l1 | 1.3 | 201 | 64 | 10 | 2 |

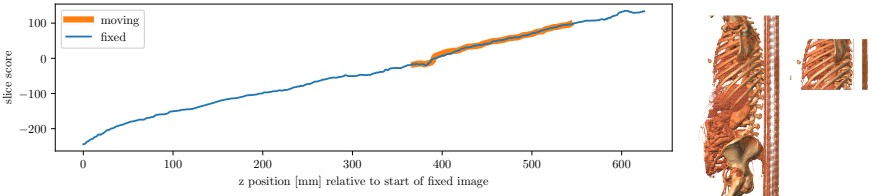

Figure 2: Prealignment according to $\ell_1$ minimization (different image pair)

faster processing, the moving image is resampled to a maximal image size of $128 \times 128 \times 128$. The fixed image is resampled to the same image resolution as the moving image. For the grid generation, we choose a sampling rate of 3, 3, and 71 in x, y, and z-direction respectively. For evaluation, we visually score the registration results into 4 categories: 1: very good alignment, 2: good alignment, 3: correct body region, and 4: failure.

## 3. Results and Conclusion

The scoring results are shown in table 1. On an NVIDIA GTX 1080 Ti, the regressor only requires 0.86 ms per slice, whereas our application needed an additional 2.9 ms to load and preprocess the DICOM slices. The resulting scores of each volume had an average Pearson correlation coefficient of 99.34% against the respective slice numbers. Figures 1 and 2 show example alignments of the score curves on two randomly selected image pairs using the two proposed methods. The $\ell_1$-based alignment gives much more stable results, at the expense of having to run all slices through the regressor. Still, the runtime is around 1 s for typical image pairs (compared to 10 ms for the fast method and about 5 seconds for FASTA).

**Conclusion:** Our SSBR-based prealignment methods are faster than FASTA and the $\ell_1$-based alignment shows also better scoring results. However, they only deliver an alignment in $z$ direction (still, the most important component when registering two axial CT scans). The $\ell_1$-based alignment is is very robust, and while it has to score all slices, the subsequent step just has to align two small 1-dimensional score arrays. For extremely fast alignment, we can just score three slices, but the resulting estimates show less precision in some cases. Both proposed methods are much more robust than traditional methods when the overlap of the volumes to be registered is small. Using the SSBR, it is even possible to align non-overlapping volumes. We plan to further evaluate this new prealignment in practice and on a larger dataset.

## Acknowledgments

We thank Bram van Ginneken and the DIAG group (Radboud UMC, Nijmegen) for making the CT data available within our common "Automation in Medical Imaging" Fraunhofer ICON project.

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
