# OpenReview forum: "Efficient Prealignment of CT Scans for Registration through a Bodypart Regressor"
_MIDL.io/2019/Conference/Abstract — MIDL Abstract 2019_

### Official Review · AnonReviewer1 · 2019-04-26
**motivation is interesting but method has weaknesses**

**Rating:** 2
**Confidence:** 3

**Review:**

This abstract uses a self-supervised bodypart regressor (Yan et al., 2018) and compares it with a grid search method for slice-wise pre-alignment of CT scans on the z-axis. Th authors motivate this work for the use for e.g. data mining in large PACS systems.

This abstract touches on an interesting application but has several weaknesses:
- the method is not novel
- the derived score is not critically evaluates, e.g. what happens in real (messy) PACS scenarios, when images are wrongly categorised, show major pathologies or artefacts from devices, have wrongly labelled modalities etc. . It is not clear if this method is robust enough for the motivated use case.
-  It is not clear if the DNN method is significantly better than the naive baseline. Also I would assume that a simple histogram-based method would also achieve reasonable results in a clean image database.
- When already using deep networks, why not predicting the z-coordinate directly from the slices, e.g., learned from a large number of aligned scans?

Overall the data-mining motivation is interesting, but there would be more potential for discussion if this would have been tried on a really messy clinical database, also containing mis-categorised modalities and artefacts.

---

### Official Review · AnonReviewer2 · 2019-05-01
**Bodypart regressor used for coarse pre-alignment of CT scans**

**Rating:** 3
**Confidence:** 2

**Review:**

The authors use the self-supervised bodypart regressor method by Yan et al for coarse prealignment of CT scans. This is achieved by aligning the regression scores of the fixed and the moving image.

Pros:
* The method is faster than the baseline (which is a grid search).
* A nice feature of the method is that can align even non-overlapping volumes.

Weaknesses:
* The evaluation is quite subjective - the scores are visually categorized into categories such as very good, good, failure, etc.
* It seems like the baseline method could perform better on a finer grid. A sampling rate of 71 units in the z-direction seems to be too coarse when the total image size is 300 slices.

Overall, I recommend acceptance as the idea of using the regressor for coarse alignment is neat.

---

### Decision · Program_Chairs · 2019-05-06
**Acceptance Decision**

Accept